# Parkinson’s Disease Multimodal Complex Treatment (PD-MCT): Analysis of Therapeutic Effects and Predictors for Improvement

**DOI:** 10.3390/jcm9061874

**Published:** 2020-06-16

**Authors:** Elke Hartelt, Raphael Scherbaum, Manuel Kinkel, Ralf Gold, Siegfried Muhlack, Lars Tönges

**Affiliations:** 1Department of Neurology, St. Josef-Hospital, Ruhr-University Bochum, 44791 Bochum, Germany; elke.hartelt@rub.de (E.H.); raphael.scherbaum@rub.de (R.S.); ralf.gold@rub.de (R.G.); siegfried.muhlack@rub.de (S.M.); 2Psychiatrisches Gutachtenbüro, 44795 Bochum, Germany; manuel.kinkel@gmx.de; 3Neurodegeneration Research, Protein Research Unit Ruhr (PURE), Ruhr-University Bochum, 44801 Bochum, Germany

**Keywords:** Parkinson’s disease, multimodal complex treatment, motor symptoms, motor complications, MDS-UPDRS Part III

## Abstract

Parkinson’s disease Multimodal Complex Treatment (PD-MCT) is a multidisciplinary inpatient treatment approach that has been demonstrated to improve motor function and quality of life in patients with Parkinson’s disease (PD). In this study, we assessed the efficacy of PD-MCT and calculated predictors for improvement. We performed a prospective analysis in a non-randomized, open-label observational patient cohort. Study examinations were done at baseline (BL), at discharge after two-weeks of inpatient treatment (DC) and at a six-week follow-up examination (FU). Besides Movement Disorders Society Unified Parkinson’s Disease Rating Scale (MDS-UPDRS) III as a primary outcome, motor performance was measured by the Timed Up-and-Go (TUG), the Berg Balance Scale (BBS) and the Perdue Pegboard Test (PPT). Until DC, motor performance improved significantly in several parameters and was largely maintained until FU (MDS-UPDRS III BL-to-DC: −4.7 ± 1.2 (SE) *p* = 0.0012, BL-to-FU: −6.1 ± 1.3 *p* = 0.0001; TUG BL-to-DC: −2.5 ± 0.9 *p* = 0.015, BL-to-FU: 2.4 ± 0.9 *p* = 0.027; *BBS* BL-to-DC: 2.4 ± 0.7 *p* = 0.003, BL-to-FU: 1.3 ± 0.7 *p* = 0.176, PPT BL-to-DC: 3.0 ± 0.5 *p* = 0.000004, BL-to-FU: 1.7 ± 0.7 *p* = 0.059). Overall, nontremor items were more therapy responsive than tremor items. Motor complications evaluated with MDS-UPDRS IV occurred significantly less frequent at DC (−1.8 ± 0.5 *p* = 0.002). Predictor analyses revealed an influence of initial motor impairment and disease severity on the treatment response in different motor aspects. In summary, we demonstrate a significant positive treatment effect of PD-MCT on motor function of PD patients which can be maintained in several parameters for an extended time period of six weeks and identify predictors for an improvement of motor function.

## 1. Introduction

Parkinson’s disease (PD) is the second-most common neurodegenerative disorder, which is neuropathologically characterized by dopaminergic dysfunction because of dopaminergic neuronal cell and axonal loss [1,2]. The first steps of the diagnosis are performed clinically and focus on motor symptoms such as bradykinesia, rigidity, tremor and postural instability [1]. In addition, various non-motor symptoms such as sleep disorders, depression, apathy, cognitive impairment and autonomic dysfunction arise [3], which substantially contribute to the reduction in patients’ quality of life [4,5]. Since there is still no curative treatment among the various therapeutic strategies available, PD remains a progressive disorder that ultimately leads to severe disability [1].

PD patients are physically less active than comparable healthy controls [6]. A higher self-reported physical activity is shown to be associated with less severe disease progression [7,8]. This is why an increased level of physical activity is recommended for this patient population and why it is claimed to have a disease modifying effect with attenuation of the progression in motor scores [9,10]. Interestingly, positive effects on motor performance have been described after completion of intense physical treatment that can persist for several months [11].

Motor and physical activity are central treatment components of the PD inpatient multidisciplinary treatment concept (Parkinson’s disease Multimodal Complex Treatment (PD-MCT)), which was established in Germany in 2008 [12]. In general, multidisciplinary approaches aim to address a broad range of various disease associated aspects. Motor symptoms, non-motor symptoms and motor complications are detailed as well as patients’ social backgrounds including the caregiver situation. The case and discharge management ensures an adequate provision of aids and the caregiver is involved in treatment decisions based on individual needs. Multidisciplinary settings include, e.g., physiotherapy, occupational therapy, speech and language therapy, specially trained nurse care and neuropsychological testing as well as possible pharmacological adjustments [13]. This approach has become a core element of PD management and has been shown to have a beneficial impact on motor and non-motor symptoms but also on quality of life [14,15]. Recent studies demonstrated that the inclusion of specialized therapists can prevent major disease complications and hospital admissions and can help to reduce overall disease costs [16]. Two randomized controlled trials showed positive effects on the disease related quality of life in PD patients [17,18].

In this manuscript, we present a prospective observational study to analyze the effects of the two-week inpatient multimodal complex treatment (PD-MCT) on motor symptoms and motor complications. Furthermore, we calculated predictors for the odds of a beneficial motor development after therapy. In this way, we aimed to identify candidates before the initiation of inpatient PD-MCT and select the patients who are most likely to benefit from this treatment.

## 2. Methods

### 2.1. Study Design and Participants

The study was designed as a prospective, non-randomized, open-label observational survey and was performed as a clinical explorative study with a planned sample size of 50 patients that were treated from November 2017 to September 2018 in the St. Josef-Hospital Bochum, Ruhr-University Bochum. All subjects gave their informed consent for inclusion before participating in the study. The Ruhr University Institutional Review Board approved the study protocol (Reg. Nr. 17-6119), which was also listed in the German Clinical Trials Register (DRKS-ID: DRKS00013361).

### 2.2. Inclusion and Exclusion Criteria, Procedures

The study workflow started with screening patients with Parkinson’s disease (PD) for eligibility who were scheduled to receive Parkinson’s disease Multimodal Complex Treatment (PD-MCT). Inclusion criteria were a diagnosis of Parkinson’s disease based on the UK Brain Bank Criteria [19] and on the International Parkinson and Movement Disorder Society (MDS) criteria [4]. Patients with other or atypical Parkinson syndromes as well as patients who were not able to take part in scoring procedures or treatment interventions due to severe mental or medical conditions were excluded. Patients’ medical histories were carefully taken and clinical examinations were performed at three different times of measurement. The baseline (BL) assessment took place on days one to three of the inpatient stay and treatment initiation. The second assessment was performed at discharge (DC), days 13 to 16. After six more weeks, patients were scheduled for an outpatient follow-up examination (FU) at the Ambulance for Movement Disorders (Figure 1).

### 2.3. Parkinson’s Disease Multimodal Complex Treatment (PD-MCT)

PD-MCT includes an interprofessional treatment carried out by physiotherapists, occupational therapists, speech and language therapists, psychologists, specially trained Parkinson nurses and neurologists. An adjustment of pharmacological treatment can also be performed. PD-MCT is integrated into the German health insurance system and thus has to comply with determined requirements. Those define a minimum of 7.5 h of treatment per week and weekly team meetings with all therapists and physicians involved [20]. The inpatient stay in our study lasted at least 14 days. During this time, patients participated in individual therapy sessions of physiotherapy, occupational therapy, speech and language therapy as well as massages, thermotherapy and group exercise. The therapy program was individually planned for each patient based on the patients’ needs as evaluated by the treating neurologist. Key elements of the applied training sessions included: gait analysis and training, balance training, fall prevention, amplitude training, posture correction, instructions for self-executed training at home, general strengthening (with or without equipment), cardiopulmonary endurance training, training of everyday life-oriented activities, development of supportive and compensation strategies, training of fine motor skills of the upper extremity, advice on medical aids, swallowing and speech training, articulation training, language cognition and psychosocial support. A detailed description on therapy components has been placed in the Appendix A. An overview about the intensity of therapeutic applications during inpatient PD-MCT is displayed in Appendix A.

### 2.4. Outcome Criteria

The main clinical outcome criteria consisted in the evaluation of motor skills, non-motor function and quality of life [15,21]. In order to stage disease severity, the modified Hoehn and Yahr scale was applied [22,23]. Motor assessments included the Movement Disorder Society’s revised version of the Unified Parkinson’s Disease Rating Scale (MDS-UPDRS) Part III [24], which was performed by certified examiners [25]. Tremor and nontremor aspects of the scale were analyzed separately [26]. Clinical phenotypes were differentiated in tremor dominant (TD) and postural instability/gait difficulty (PIGD) [27]. Further scores for motor function were the Timed Up and Go Test (TUG) [28,29] and the Berg Balance Scale (BBS) [30,31]. Since one clinical rating scale alone does not sufficiently display the impact on gait, balance and posture due to PD, we combined those scales for a detailed evaluation of motor impairment in our patients [32]. Both tests were performed by professional members of the physiotherapy team. Additionally, the Perdue Pegboard Test (PPT) [33,34] was carried out as part of the occupational therapy setting. Further parameters were the MDS-UPDRS Part I (non-motor experiences of daily living) [35], MDS-UPDRS Part II (motor experiences of daily living) [36] and MDS-UPDRS Part IV (motor complications). Cognitive impairment was evaluated with the Montreal Cognitive Assessment (MoCA) [37,38]. To estimate the change as an overall treatment effect, the patient’s Clinical Global Impression of Change (pCGI-C) was recorded twice, at DC and at FU [39,40]. Other outcome criteria focused on non-motor symptoms such as apathy and depression. For this purpose, we assessed the Apathy Evaluation Scale in German translation (AES-D) [41], the revised Beck-Depressions Inventory (BDI-II) [42,43] and a shortened version of the Hamilton Depression Rating Scale (HAMD-17) [44] at all three assessment times. Questionnaires and scores were partly explored in interview settings, filled out by the patient him- or herself or answered assisted by the caregiver. For the BL characterization of our study population, we implemented several clinical scores.

### 2.5. Statistical Analysis

Main outcome criteria were visualized using graphical descriptive methods such as boxplots, scatter plots and Q-Q plots identifying outliners and indicating normality for further analyses. Changes over time of the outcome parameters were analyzed using repeated measures analysis of variance (rmANOVA) and post hoc analyses. Correlations between the variables were assessed with Spearman’s rank correlation coefficient. Odds ratios and relative risks were used on dichotomized variables to identify possible predictors of treatment response. In order to create dichotomized variables of the clinical outcome parameters, we followed the recommendations of Martinez-Martin for grading the MDS-UPDRS [45]. In this context, some patients were classified differently from their original severity level. In MDS-UPDRS part I and II, one patient was allocated to “moderate impairment” although they formally belonged to the group “severe.” In MDS-UPDRS parts III and IV, this was the case for two patients in each part, respectively. The significance level was determined at 5% as well as at 1% in order to reduce type I error probability. All data were analyzed by means of IBM SPSS Statistics version 25.

## 3. Results

Of all 69 patients screened in the assessment period, 47 patients met the inclusion criteria and were assessed at BL. After two weeks of inpatient treatment, 43 patients completed the PD-MCT and were assessed at DC. The FU examination could be performed only on 38 patients because of patient dropouts in five cases, mostly due to the withdrawal of consent for continuation (Figure 2).

The characterization of the study population according to clinical characteristics and demographics at BL is shown in Table 1 and has been presented with different focus in a previous analysis [15]. The mean patient age was 68.5 (SD ± 9) years and there were no relevant differences in the female/male ratio. Almost half of the patient population presented with postural instability (Hoehn and Yahr ≥ 3) or had motor complications as documented in the MDS-UPDRS part IV (55.3%). The predominant motor phenotype consisted of the postural instability/gait difficulty (PIGD) variant in our population (PIGD 59.6%), whereas tremor dominant (TD) phenotypes were 25.5% and indeterminate (ID) were only 14.9% [27] of the cases. Two-thirds of the patients had at least mild cognitive impairment as evaluated by the MoCA [15].

### 3.1. Significant Correlation between Motor Parameters

Correlation analyses applying Spearman’s rank correlation showed significant results between MDS-UPDRS part III and other motor parameters. Both MDS-UPDRS II and Timed Up-and-Go Test (TUG) strongly correlated with MDS-UPDRS III at BL (MDS-UPDRS-II r_s_ = 0.620 *p* = 0.000003, TUG r_s_ = 0.483 *p* = 0.001) and at DC (MDS-UPDRS II r_s_ = 0.668 *p* = 0.000, TUG r_s_ = 0.417 *p* = 0.006) (Figure 3A,B). Strong associations between the Berg Balance Scale (BBS) and the Purdue Pegboard Test (PPT) with the MDS-UPDRS III were also noted. However, these values presented as negative correlations because by definition good motor performance led to increases in BBS and PPT scores, while MDS-UPDRS III motor scale values decreased with reduced motor symptom burden (BL-BBS r_s_ = −0.417 *p* = 0.005, DC-BBS r_s_ = −0.523 *p* = 0.0004, BL-PPT r_s_ = −0.591 *p* = 0.00002, DC-PPT r_s_ = −0.491 *p* = 0.001) (Figure 3C,D). Interestingly, the relations between BL scores (green) remained overall stable at DC (red) because all motor parameters improved during inpatient PD-MCT (Figure 3).

With respect to the performed interventions during the PD-MCT, we analyzed the relation between the amount of therapy hours applied during PD-MCT and the development of distinct motor parameters until discharge as shown in Appendix A. Interestingly, the results display a significant correlation between the amount of physiotherapy and the improvement in the PPT (r_s_ = −0.352, *p* = 0.028). In addition, a significant correlation between occupational therapy and an improvement in TUG could be shown (r_s_ = 0.373, *p* = 0.019). Regarding adjustment of medication, an increase in overall LED correlated with an improvement in MDS-UPDRS II (r_s_ = −0.35, *p* = 0.018). Separate analyses of MDS-UPDRS III tremor and nontremor subgroups [26] revealed that most significant correlations can be attributed to the impact of nontremor items. Further details on correlation analyses are presented in the Appendix A.

### 3.2. Improvement of Distinct Clinical Outcome Parameters over Time

Non-motor aspects as well as motor aspects of daily living improved significantly at DC (MDS-UPDRS I mean difference −3.2 points ±0.6 SE, *p* = 0.0004; MDS-UPDRS II −2.2 ± 0.6 *p* = 0.003) (Figure 4). Motor symptoms, as evaluated by MDS-UPDRS part III, improved by −4.7 ± 1.2 points at DC (p = 0.0012) and by −6.1 ± 1.3 points at FU (*p* = 0.0001). Analyzing the tremor and nontremor parts of MDS-UPDRS III separately [26], the sustaining significant improvements of motor function can be attributed primarily to nontremor items (tremor items DC: −0.8 ± 0.4 points *p* = 0.216, FU: −0.7 ± 0.5 points *p* = 0.561; nontremor items DC: −3.9 ± 1.2 points *p* = 0.005, FU −5.4 ± 1.1 points *p* = 0.00005). The Timed Up-and-Go Test also showed a persistent improvement (DC: −2.5 ± 0.9 s *p* = 0.015, FU: 2.4 ± 0.9 s *p* = 0.027). The Berg Balance Scale and the Purdue Pegboard Test improved significantly until DC (BBS 2.4 ± 0.7 points *p* = 0.003, PPT 3.0 ± 0.5 sticks *p* = 0.000004), which did not maintain until FU [15]. Concerning motor complications, the significant improvement of MDS-UPDRS IV occurred with a relative change of −38.1% at DC (*p* = 0.002) (Figure 4). The extent of training activity in outpatient care until FU was individually different and was not performed in a standardized setting, so that no specific analysis could be performed. The type of care facility after DC was primarily home care (Table 2).

### 3.3. Prediction of Treatment Response in Relation to Motor and Non-Motor Function at BL

In order to identify possible predictors for treatment response, odds ratios and relative risks were calculated for dichotomized outcome parameters and predictor variables (Appendix A). The purpose was to explore whether defined disease parameters are related to treatment responses as measured by MDS-UPDRS parts I–IV, TUG, BBS and PPT at DC. Responders were defined to have at least one score improved from BL assessment to DC (ΔBL-DC < 0).

The odds of motor improvement in MDS-UPDRS part III were 0.13-fold lower for mildly motor impaired patients (MDS-UPDRS III score < 33 points, *p* = 0.004) compared to moderately impaired ones. Similarly, the odds of balance improvement as measured by BBS were 0.12-fold lower for patients with Hoehn and Yahr stages < 3 (*p* = 0.002). Regarding motor complications, patients with no or only little impairment at BL (MDS-UPDRS IV <5 points) had 0.13-fold lower odds to benefit compared to more affected individuals (*p* = 0.005). Under a significance level of 0.05, there were further putative predictors identified, such as a disease duration <8 years and a daily levodopa equivalent dose (LED) of <595 mg led to 4.75-fold higher odds to show an improvement in motor aspects of daily living (MDS-UPDRS part II). A normal cognition with MoCA scores ≥ 26 showed 0.23-fold lower odds for motor response in MDS-UPDRS part III than with mild cognitive impairment (MCI) [15]. Concerning motor response in the TUG, having less than 3.5 h of occupational therapy per week led to 4.5-fold higher odds to profit.

## 4. Discussion

The results of our study pointed out several beneficial effects of the two-week multimodal complex treatment (PD-MCT) on motor aspects in patients with PD. Overall, we could demonstrate that motor performance improved significantly in patients subjected to PD-MCT and that the severity of motor complications was reduced. Motor parameters at BL and at DC strongly correlated with each other, which indicates a stable link between different domains of motor symptoms in our study population. Interestingly, associations between distinct BL characteristics and the odds of motor improvement at DC were identified, which point out putative predictors of treatment response.

The detailed assessment of motor manifestations using MDS-UPDRS part III, TUG, BBS and PPT represents a well-established and reliable combination of scores in clinical PD investigations [46,47]. We found a significant improvement of the MDS-UPDRS III score which was maintained until follow-up [15]. Interestingly, the beneficial effect of PD-MCT mostly comprises nontremor symptoms. This, however, is relevant for the majority of our study population because it presented with a nontremor dominant phenotype. The improvements in motor function could only partially be maintained until FU. Two other studies with more intense or longer treatment duration did not demonstrate a significant decline in motor function at FU [18,48]. A reason for the limited preservation in our study population could be that only 28.9% were regular-exercisers with an activity level of >150 min/week in everyday-life, whereas the larger proportion was low or non-exercisers. However, there was no significant connection identified between the demonstrated improvement at FU and the outpatient activity level.

Regarding MDS-UPDRS III predictor analysis, it was revealed that motor benefit was more likely to occur in patients with at least moderate motor impairment (Appendix A) [15]. This is underlined by the relevant mean difference in MDS-UPDRS III responders and non-responders at DC regarding the BL motor level (Appendix A). These findings suggest that patients who are rather mildly impaired as measured by the MDS-UPDRS III are not the strongest benefitting group from the PD-MCT concerning overall MDS-UPDRS III function. In order to decide which patient characteristics are preferable for a beneficial participation, more aspects have to be taken into consideration. The motor experiences of daily living (MDS-UPDRS II), representing the overall level of motor disability [36], showed a significant short-term improvement. However, this improvement does not represent a clinically relevant amount, as a total change of −2.2 points is below the minimal clinically important difference (MCID) of 3.05 points [49]. In this context, a disease duration <8 years and a LED intake of <595 mg/day seemed to predict higher odds to profit considering the level of motor disability. Half of the patients in our study were classified as Hoehn and Yahr ≥ 3. This group of patients, presenting with postural instability, was more likely to profit in balance performance as measured by BBS than lower staged patients. The population showed a mean age of 68.5 years, a mean disease duration of 8.5 years and a mean motor impairment of 37.5 points in MDS-UPDRS part III at admission, mostly defined by the 23 nontremor items. Therefore, we consider our results most valid for at least moderately impaired patients. The amount of applied therapy hours of PT and OT showed a significant correlation with the improvement in PPT and TUG scores. Since we could show a strong correlation within the motor scores, a distinct relation between applied interventions and motor improvement can be assumed.

More than half of our study population suffered from motor complications at BL analysis as documented in the MDS-UPDRS IV. This is a relatively high prevalence compared to frequency analyses that reported levodopa-induced dyskinesia of around 40% after four to six years of treatment [50]. The MDS-UPDRS IV does not distinguish between different types of dyskinesia such as wearing-off hypokinesia or painful off-state dystonia but is a frequently used scale for motor complications [24,51]. In other publications, it was suggested to carefully differentiate between peak-dose dyskinesia, diphasic dyskinesia and wearing-off hypokinesia in order to identify patients’ specific needs of treatment [52]. Some trials state the appearance of dyskinesia, commonly referred to as levodopa-induced dyskinesia (LID), to be more likely in akinetic-rigid phenotypes, than in tremor-dominant ones [53]. This fits the characteristics of our study population with only 26% of the patients categorized as tremor-dominant phenotype demonstrating a high frequency of motor complications. Moreover, it has been shown previously, that patients with motor complications are more likely to be subjected to PD-MCT [54]. Interestingly, our data showed a temporary, statistically significant improvement of motor complications by 38% or 1.8 points in MDS-UPDRS IV from BL to DC in PD-MCT treated patients. This improvement is higher than the threshold for MCID of 0.9 points and thus should be interpreted as a clinically meaningful development [55]. Apart from the MDS-UPDRS IV, a more detailed and comprehensive score was developed to evaluate the impairment and disability resulting from involuntary movements, the Unified Dyskinesia Rating Scale (UDysRS) [56]. Since the scale depicts the impact of dyskinesia on activities of daily living and includes subjective as well as objective dyskinesia rating, it might be superior to the MDS-UPDRS IV in detecting treatment effects [57]. The UDysRS is frequently used in publications presenting positive effects of pharmacological agents as Amantadine on dyskinesia [58,59,60].

Major and disabling dyskinesia seems to be predicted by longer disease duration, younger age at onset and higher levodopa-dose rather than early levodopa initiation [52,61,62]. Our data similarly pointed out significant correlations between motor complications and the age at diagnosis, the disease duration and daily levodopa equivalence dose. A greater non-motor burden in MDS-UPDRS I, BDI-II or AES was not identified to correlate with or predict higher odds of motor complication occurrence [51]. Even though studies revealed that PD patients either tend to prefer dyskinesia over symptoms of hypokinesia [63] or can be unaware of mild dyskinesia [51], motor complications are a common problem, which can significantly affect patients’ quality of life [64,65]. There are different approaches evaluating motor complications in PD patients. The assessment of its influence on patients’ quality of life can be difficult because of inconsistent documentation of occurrence and severity [66]. For that purpose, it is necessary to assess and differentiate characteristics of motor fluctuations by consistent and patient independent methods in addition to the MDS-UPDRS IV. Here, the use of wearable sensors as an objective and continuous monitoring of dyskinesia should be taken into consideration [67,68,69].

There are some additional limitations which need to be mentioned for this pilot study. Due to the observational study design with lack of a control group, a causal relation between the multimodal complex treatment and the described developments cannot be claimed. The selection of statistical analyses was limited by the small sample size and heterogeneous study population. In general, it is difficult to determine the impact of single treatment components on defined outcome criteria in multidisciplinary or complex treatment settings because there is a huge variety of interventions performed [70]. In addition, PD-MCT also permits an adjustment in the patients’ PD medication if indicated. Thus, patients with predominantly parkinsonian hypokinesia were adjusted with a total LED increase (e.g., higher levodopa dose including dopamine receptor agonists, increase of daily intake frequencies, administration of additional drugs such as MAOB-inhibitors or COMT-inhibitors), whereas patients with LID received an overall dose reduction. In total, there was an increase in daily LED from BL to DC of 16% with a heterogenous intensity of optimization in the patient population [15]. Correlation analyses revealed a significant relation between LED increase at DC and the improvement in MDS-UPDRS II. Surprisingly, no connection between LED increase and the change in MDS-UPDRS III or IV could be shown, although the impact of medical treatment is expected to occur faster than complementary treatment effects. However, overall disease progression is claimed to remain widely unaffected by pharmacological treatment [71].

We hold the view that the positive impact of PD-MCT on several motor and non-motor symptoms after inpatient treatment does not only represent an immediate and short-time effect. It is expected to improve motor performance and quality of life even for a much longer period serving as an initial facilitator and change stimulus. The 14-day inpatient stay aims to point out the patients’ deficits in their motor and non-motor function which are most promising to be improved in their everyday life. Moreover, the overall awareness of symptoms and dysfunctional behavior is stressed. Through individual education and instructions on regular self-executed and supported exercises, patients are motivated to adjust their daily routine back at home. From other studies, e.g., as implemented in the Dutch ParkinsonNET [72], we have learned that an integrative care network with specialized physiotherapy concepts for PD patients leads to fewer disease related complications in later phases of disease [73]. The innovative approach depicts the importance of a closely linked inpatient and outpatient treatment sector in order to be beneficial to patients’ quality of life [74,75].

We propose a follow-up study that further explores the potential of the well-established PD-MCT in a multi-center setting. In addition, a control-treatment group without inpatient PD-MCT would have to be added and a more longitudinal FU should be included. This would also enable to identify patients with PD even better who need inpatient treatment or who could benefit similarly well from an outpatient-based treatment regime.

## Figures and Tables

**Figure 1 jcm-09-01874-f001:**
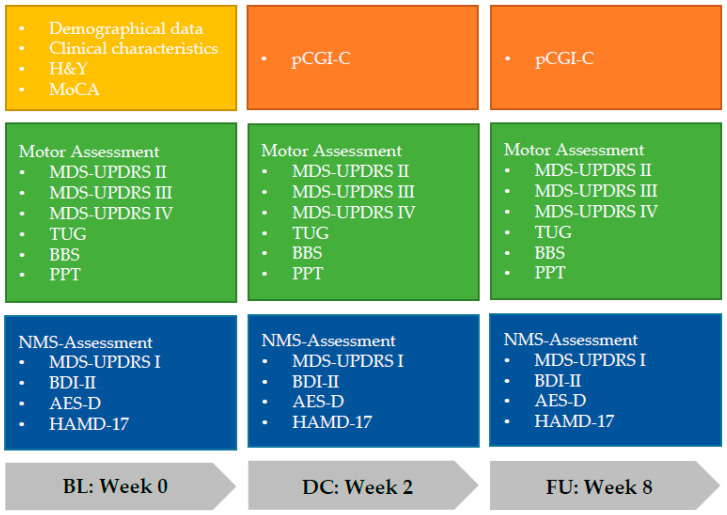
Type and Timing of Clinical Assessments. H&Y: modified Hoehn and Yahr stage, MoCA: Montreal Cognitive Assessment, MDS-UPDRS: Movement Disorders Society Unified Parkinson’s Disease Rating Scale, Part I–IV, TUG: Timed Up and Go Test, BBS: Berg Balance Scale, PPT: Purdue Pegboard Test, NMS: Non-Motor Symptom, BDI-II: revised Beck Depression Inventory, AES: Apathy Evaluation Scale, HAMD-17: Hamilton Rating Scale for Depression, pCGI-C: Patient’s Clinical Global Impression of Change, BL: Baseline, DC: Discharge, FU: Follow-Up.

**Figure 2 jcm-09-01874-f002:**
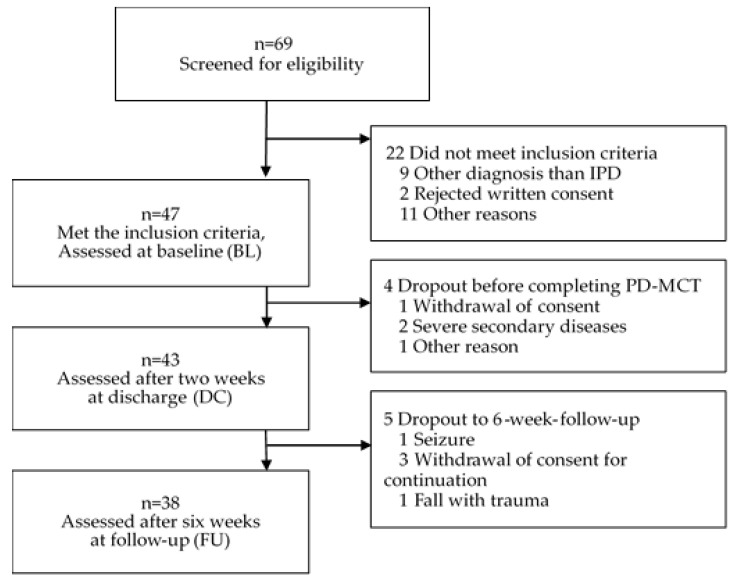
Study Flow Diagram. Data collecting period from November 2017–September 2018.

**Figure 3 jcm-09-01874-f003:**
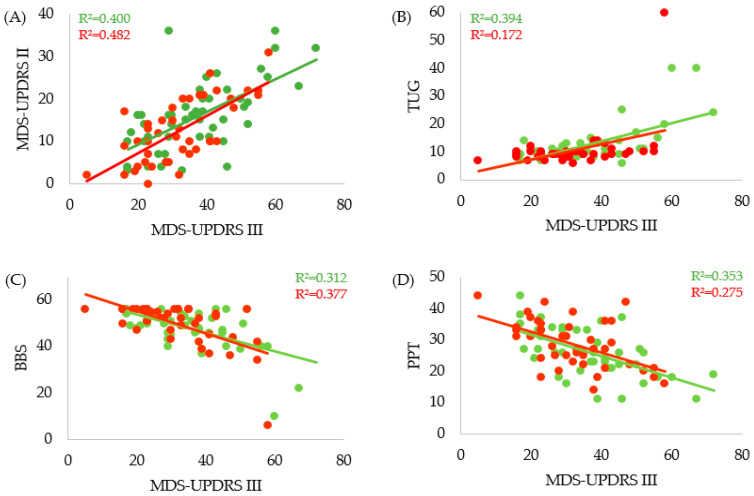
Significant correlations between MDS-UPDRS Part III and distinct motor parameters at BL (green) and DC (red). Trend lines indicate a linear relation. Significance level at 5%. (**A**) Correlation of MDS-UPDRS II and MDS-UPDRS III. (**B**) Correlation of TUG and MDS-UPDR III. (**C**) Correlation of BBS and MDS-UPDRS III. (**D**) Correlation of PPT and MDS-UPDRS III. Parameters are scaled in absolute counts of each scale, except for TUG which is measured in seconds. MDS-UPDRS: Movement Disorders Society Unified Parkinson’s disease Rating Scale Parts II and III, TUG: Timed Up and Go, BBS: Berg Balance Scale, PPT: Purdue Pegboard Test.

**Figure 4 jcm-09-01874-f004:**
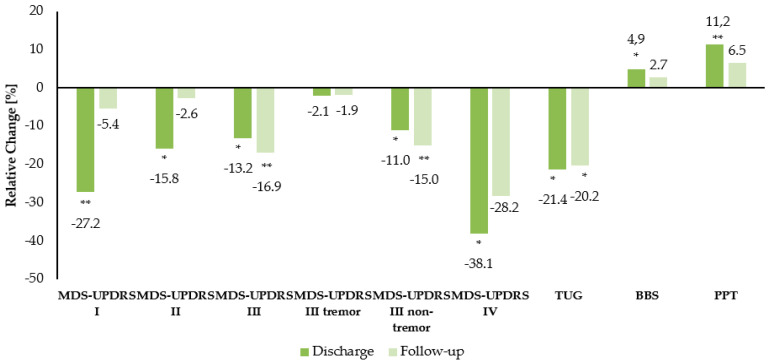
Relative change in distinct clinical outcome parameters from baseline to discharge (DC) and at a six-week follow-up (FU) illustrated as columns in %. Dark columns represent relative change at DC, light columns represent relative change at six-week FU. A *p*-value < 0.05 in post hoc analyses of repeated measures ANOVA is indicated by one asterisk (*), a *p*-value < 0.001 is indicated by two asterisks (**). MDS-UPDRS Movement Disorder Society Unified Parkinson’s Disease Rating Scale, Parts I–IV, TUG Timed Up and Go, BBS Berg Balance Scale, PPT Perdue Pegboard Test.

**Table 1 jcm-09-01874-t001:** Demographics and clinical characteristics at baseline.

Variable	Value	
Age, years	68.5	(9.0)
Female/male sex	22/25	(46.8/53.2)
Disease duration, years	8.5	(5.3)
H&Y, median (IQR)	3	(2.5–3)
<3	23	(48.9)
≥3	24	(51.1)
TD Phenotype	12	(25.5)
PIGD Phenotype	28	(59.6)
ID Phenotype	7	(14.9)
Motor Fluctuations, yes/no, n = 46	26/20	(55.3/42.6)
Daily LED, mg	698.3	(446.4)
MDS-UPDRS I	12.6	(5.9)
MDS-UPDRS II	16.1	(8.4)
MDS-UPDRS III	37.5	(13.8)
Tremor items	7.0	(4.5)
Nontremor items	30.4	(12.0)
MDS-UPDRS IV	5.2	(4.0)
TUG, n = 45	12.8	(7.1)
BBS, n = 44	47.6	(9.1)
PPT, n = 44	26.0	(7.7)
BDI-II, n = 46	11.9	(6.7)
HAMD-17, n = 46	7.4	(3.5)
AES, n = 46	8.4	(6.2)
MoCA	22.5	(4.6)

If not indicated otherwise, data are presented as mean (SD) or n (%). N = 47 if not indicated otherwise. H&Y: modified Hoehn and Yahr stage, IQR: interquartile range, TD: tremor dominant, PIGD: postural instability/gait difficulty, ID: indeterminate, MDS-UPDRS: Movement Disorders Society Unified Parkinson’s disease Rating Scale, Part I–IV, TUG: Timed Up and Go Test, PPT: Purdue Pegboard Test, BBS: Berg Balance Scale, BDI-II: revised Beck Depression Inventory, HAMD-17: Hamilton Rating Scale for Depression, AES: Apathy Evaluation Scale, MoCA: Montreal Cognitive Assessment, LED: Levodopa equivalent dose [15].

**Table 2 jcm-09-01874-t002:** Outpatient activity level and discharge facility.

Therapy Intervention	Value
Outpatient activity, n = 38	
Non-exercisers	3 (7.9)
Low-exercisers	24 (63.2)
Regular-exercisers	11 (28.9)
Discharge facility, n = 47	
Home	44 (93.6)
Surgery department	1 (2.1)
Geriatric rehabilitation	2 (4.3)

Values presented as total (%). Outpatient activity level classified as non-exercisers with no physical activity at all, low exercisers with 1–150 min/week and regular-exercisers with >150 min of physical activity per week [8].

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
