# Peer review of "Parkinson’s Disease Multimodal Complex Treatment (PD-MCT): Analysis of Therapeutic Effects and Predictors for Improvement"

_jcm, 2020, doi:10.3390/jcm9061874_

Round 1

Reviewer 1 Report

Overall, this paper is well written and provides further support for the specific multimodal treatment for PD discussed. Here are my comments:

Throughout the paper, the treatment protocol is referred to as the Parkinson’s Disease Multimodal Complex Treatment (PD-MCT) yet the article title refers to it differently (MCT-PD). If this is a specific, standard protocol, then the title should be consistent with the rest of the paper.

Line 53: change to “was established in Germany…”

Line 59: grammatical error

Line 86: how was “severe mental” conditions determined? Was this by use of the MoCa and if so, what cut-off score was used to determine this?

Line 125: I do not get a clear picture of what exactly PD-MCT truly entails. Do all participants receive all the disciplines mentioned? Or is this based on assessment?  The authors state a minimum of 7.5 hours a week – for this study, what was the average weekly treatment time? Is this time shared with all the disciplines mentioned? It would be nice to see a breakdown of how many actual hours the participants got for certain therapies, such as PT, on average.

Table 2: needs units (hours, years, etc)

Line 178: add “with” in between “presented” and “postural instability”

Line 218: The results of MDS-UPDRS IV is reported as a relative change percentage –which differs from the reporting of all the other variables. What is the mean difference?

Lines 280 – 282: There is a discrepancy in the score reporting here. It states that the MDS-UPDRS II improved by a score of 15.8. However, based on Figure 4, this represents a % relative change and not the actual score. Line 212 reports a score change of -2.2 points, which would not go beyond the MCID of -3.05 points.

Lines 337-339: while it is true that overall disease progression is not widely affected by pharmacological treatment, this treatment can have improved short-term impact on motor aspects of PD. Lines 333-334 reports that those with hypokinesia were adjusted with a total LED increase. One purpose of this increase is to improve motor function, therefore I am unsure if you can completely rule out the adjustments in medications in the improvement of motor function.

Reviewer 2 Report

The authors demonstrate a significant positive treatment effect of PD-MCT on motor function of PD patients which can be maintained in several parameters for an extended time period of six 31 weeks and identify predictors for improvement of motor function.

There were certain issues for the manuscript:

  1. In general, the content of this paper was very alike what the authors just published ( J Neurol. 2020 Apr, 267(4): 954-965) with little additional information.
  2. (Fig 3) The correlation between motor performance in PD patients was straightforward and not surprising. Authors should conduct more specific analysis to identify any novel findings.
  3. The same issue as Figure 4. A multi-domain intervention resulted in a short-term improvement of PD symptoms were quite common. It is quite obvious that the effect waned during follow up, which is also usually the case. Authors should conduct more specific analysis to identify any novel findings. 
  4. In table 2, the authors set plenty of cut-off values for each task. Is there any basis for these cut-off values? For instance, please clarify the reason of setting LED at 595mg as cut-off? Is there any clinical significance?
  5. In table2 as well, in total, authors conduct nearly a hundred times of statistic comparison. In this case, it is not surprising to obtain 3-5 parameters which were p<0.05, based on the type I error of statistic comparison, the author may consider set p value<0.01 as truly statistics significance. 
  6. Minor issues, is there any reason for the PD progression in 2 cases? Did the progression result from the MCT?

Reviewer 3 Report

This manuscript shows a very interesting approach for PD patients treatment and this work allows us to have objective data in order to make a good selection of candidates and to underline the long term efficacy of interventions. I congratulate the authors for the hard work that is reflected in their submission.
In my opinion, this paper would improve if some suggestions are considered.
1. The authors make a follow-up assessment at week 8 in order to state the persistence of the multimodal therapy effects. Considering the heterogeneity of the sample and duration of the intervention (14 days), I consider it necessary to document the type of facilities where the patients were discharged to ( elder residence, own house, another hospital, etc.). In order to understand the persistence of the effects it is also necessary to know if some or any of them kept receiving any of the multimodal treatments (physiotherapy, occupational therapy, etc) as part of their ambulatory care and how many hours per week.
2. On the other hand, the 5 dropouts are described as withdrawal of consent for continuation in the main text but in Fig2. It states the following reasons: 2 PD progression; 3 Lack of compliance. As this is an in-patient treatment, I find surprising the lack of compliance stated and I consider that the dropout of patients who worsened may alter the outcomes stated. I recommend considering the inclusion of these subjects in the analysis.

Round 2

Reviewer 2 Report

I appreciate the response from the authors. I embrace most of them, however, there were still some minor questions

  1. Some improvements sustained at the FU, does the sustain relate to the discharge activity? 
  2. LED difference did not correlate with the change of either UPDRS-III or IV, which is not common since medical treatment usually take action faster than other physical training.
